# Validation of the T Descriptor (TNM-8) in T3N0 Non-Small-Cell Lung Cancer Patients; a Bicentric Cohort Analysis with Arguments for Redefinition

**DOI:** 10.3390/cancers13081812

**Published:** 2021-04-10

**Authors:** Philip Baum, Samantha Taber, Stella Erdmann, Thomas Muley, Mark Kriegsmann, Petros Christopoulos, Michael Thomas, Hauke Winter, Joachim Pfannschmidt, Martin E. Eichhorn

**Affiliations:** 1Department of Thoracic Surgery, Thoraxklinik, Heidelberg University Hospital, Roentgenstrasse 1, 69126 Heidelberg, Germany; Hauke.Winter@med.uni-heidelberg.de (H.W.); martin.eichhorn@med.uni-heidelberg.de (M.E.E.); 2Department of Thoracic Surgery, Heckeshorn Lung Clinic, Helios Klinikum Emil von Behring, Walterhöferstraße 11, 14165 Berlin, Germany; Samantha.Taber@helios-gesundheit.de (S.T.); joachim.pfannschmidt@helios-gesundheit.de (J.P.); 3Institute of Medical Biometry and Informatics, Heidelberg University Hospital, Im Neuenheimer Feld 130.3, 69120 Heidelberg, Germany; erdmann@imbi.uni-heidelberg.de; 4Translational Research Unit, Thoraxklinik, Heidelberg University Hospital, 69120 Heidelberg, Germany; Thomas.Muley@med.uni-heidelberg.de; 5Translational Lung Research Center Heidelberg (TLRC), Member of the German Center for Lung Research (DZL), 69120 Heidelberg, Germany; Mark.Kriegsmann@med.uni-heidelberg.de (M.K.); petros.christopoulos@med.uni-heidelberg.de (P.C.); michael.thomas@med.uni-heidelberg.de (M.T.); 6Institute of Pathology, Heidelberg University Hospital, 69120 Heidelberg, Germany; 7Department of Thoracic Oncology, Thoraxklinik, Heidelberg University Hospital, Roentgenstrasse 1, 69126 Heidelberg, Germany

**Keywords:** lung cancer, surgery, IASLC, T3N0, survival, chemotherapy, TNM

## Abstract

**Simple Summary:**

Lung cancer patients have different survival outcomes depending on tumor size and growth pattern after surgery. This study aims to optimize tumor classification, and identify patients who could benefit the most from additional chemotherapy after surgery. In a specific lung cancer cohort, we study how a new redefinition of tumor classification could lead to a more solid recommendation of which patients to offer chemotherapy after surgery.

**Abstract:**

The current pT3N0 category represents a heterogeneous subgroup involving tumor size, separate tumor nodes in one lobe, and locoregional growth pattern. We aim to validate outcomes according to the eighth edition of the TNM staging classification. A total of 281 patients who had undergone curative lung cancer surgery staged with TNM-7 in two German centers were retrospectively analyzed. The subtypes tumor size >7 cm and multiple nodules were grouped as T3a, and the subtypes parietal pleura invasion and mixed were grouped as T3b. We stratified survival by subtype and investigated the relative benefit of adjuvant chemotherapy according to subtype. The 5-year overall survival (OS) rates differed between the different subtypes tumor diameter >7 cm (71.5%), multiple nodules in one lobe (71.0%) (grouped as T3a), parietal pleura invasion (59.%), and mixed subtype (5-year OS 50.3%) (grouped as T3b), respectively. The cohort as a whole did not gain significant OS benefit from adjuvant chemotherapy. In contrast, adjuvant chemotherapy significantly improved OS in the T3b subgroup (logrank *p* = 0.03). This multicenter cohort analysis of pT3N0 patients identifies a new prognostic mixed subtype. Tumors >7 cm should not be moved to pT4. Patients with T3b tumors have significantly worse survival than patients with T3a tumors.

## 1. Introduction

TNM classification changes from the seventh edition (TNM-7) to the eighth edition (TNM-8) are based on current analyses of the International Association for the Study of Lung Cancer (IASLC) database [1]. In the last analysis, 2108 patients with pathologically staged tumors were classified as pT3N0, and most of these patients (79%) came from Japan, South Korea, or the People’s Republic of China [2]. The T3 category for lung cancer represents a heterogeneous group based on tumor size, separate tumor nodes in one lobe, and locoregional spread. Accordingly, the T3 descriptor was modified both in revisions to TNM-7 and to TNM-8 (Table 1). In TNM-6, tumors with a diameter larger than 3 cm were classified as T2, and satellite tumor nodules within the same lobe as the primary-tumor were classified as T4; separate metastatic tumor nodule(s) in the ipsilateral non-primary lobe(s) of the lung were classified as M1 [3]. In TNM-7, tumor size 5–7 cm was designated as T2b, and tumors >7 cm, or with separate nodules in the same lobe, were categorized as T3. Parietal pleura invasion, tumor distance of less than 2 cm from the main carina, and atelectasis or obstructive pneumonitis of the entire lung also belonged to T3 [4]. In 2015, the Staging and Prognostic Factors Committee of the IASLC published proposals to revise the lung cancer staging criteria for TNM-8 [2]. Tumors larger than 5 cm were reclassified from T2 to T3, and tumors >7 cm were reclassified as T4. Diaphragm invasion was moved to T4, and lung atelectasis, and all cases of main bronchus invasion, regardless of the distance from the carina, were defined as T2 (Table 1).

The aim of this study is to present up-to-date outcomes in surgically resected patients with pT3N0 tumors according to TNM-7 to validate the current T3 descriptors and evaluate the impact of adjuvant chemotherapy on overall survival.

Since there have been several changes within the pT3N0 category, the effect of adjuvant chemotherapy on long term survival is insufficiently understood. The current ASCO guidelines for resected non-small cell lung cancers recommend cisplatin-based chemotherapy for UICC Stage IIA/B and IIIA tumors, and grade the evidence of this recommendation as high [5]. Although adjuvant chemotherapy is considered the standard of care for resected (>Stage I) NSCLC, in light of toxicity and compliance, we need to better identify which patient subgroups are more likely to benefit from adjuvant therapy [6,7,8].

## 2. Materials and Methods

### 2.1. Patients

This is a retrospective analysis of patients treated in two tertiary high-volume centers in Germany (Heidelberg and Berlin), who were surgically resected and staged as pT3N0 lung cancer according to the TNM-7 staging classification. Clinical data of patients who underwent surgery from 1 March 2009 to 30 June 2016, were retrospectively selected from prospectively maintained databases. The Heidelberg database was approved by the institutional review board of the Heidelberg University (No. S-174/2019). In the Berlin database, all patients had given their informed consent to have their data used for research purposes, so that the institutional review board waived the need for registration. Patients from 2009 staged according to TNM-6 were reclassified in accordance with TNM-7.

Typical (*n* = 6) or atypical (*n* = 5) carcinoid tumors and small-cell or combined small-cell cancer tumors (*n* = 4) were excluded from further analysis. Patients who received neoadjuvant treatment were excluded (*n =* 31). Patients who underwent surgery but did not receive lymph node resection were not analyzed (*n =* 5). R2-resections (*n =* 7) were excluded. Cases in which pT3 was based on atelectasis (*n =* 3), tumor distance <2 cm from the carina (*n =* 9), or mediastinal pleura invasion (*n =* 5) were excluded because of very low numbers. Patients who died within 30 days of surgery (*n =* 6) were excluded from analysis. A rate of 2.1% is below average postoperative mortality rates in Germany [9]. The final study set included 281 patients (Heidelberg *n =* 164, Berlin *n =* 117).

Four major T-descriptor subtypes were present (Table 2): (1) tumor diameter >7 cm (*n =* 127, (45.2%)), (2) two or more separate nodules in the same lobe (*n =* 77, (27.4%)), (3) parietal pleural invasion (*n =* 33, (11.7%)). A fourth subtype was defined as “mixed” and included cases where criteria for two or more of the three major subtypes were present. This mixed subtype cohort consisted of 44 patients (15.7%).

### 2.2. Preoperative Staging, Surgery, and Adjuvant Therapy

Initial staging was performed with FDG-PET-CT and brain MRI in addition to the diagnostic chest computed tomography (CT). Tumor staging of all patients was based on the 7th edition of the TNM-classification system, which was valid during the complete treatment period. After staging, patients were discussed in a multidisciplinary tumor board before surgery. Systematic hilar and mediastinal lymph node dissection was performed according to standard practice [10]. Depending on postoperative recovery, constitution, and risk profile, decisions to offer adjuvant platinum-based doublet chemotherapy were made on a case-by-case basis.

### 2.3. Follow-Up and Statistical Analysis

Follow-up visits were scheduled at the outpatient service every 3 months for the first two years and subsequently every 6 months for the next 3 years. Recurrence of disease resulted in whole-body restaging and interdisciplinary discussion. Information on cancer and survival status was updated by specially trained research personnel from the authors’ institutions.

The endpoint overall survival was defined as the time from date of diagnosis to the date of death or last follow-up (in censored, still living patients). Data were collected and analyzed using Stata Version 16 (PB), and the results independently checked with R Version 4.0.2 (SE). Associations between stage pT3N0 and the other variables were tested by the chi-square test. To estimate survival, the Kaplan–Meier product limit method was used, and the survival time and the associated 95% confidence intervals were calculated using Greenwood’s formula. Two-tailed logrank-tests for the whole observation period were used to compare different groups. Cox-regression analysis was conducted to evaluate the influence of selected covariables (age, ECOG, adjuvant chemotherapy, histology, gender) on the primary endpoint. We report the median follow-up time calculated by the inverse Kaplan–Meier product limit method with corresponding 95% confidence intervals and “stability interval” as suggested by Schemper and Betensyk, respectively [11,12]. In survival graphs, censored observations are shown as ticks on the line. A *p*-value of <0.05 was considered statistically significant. However, the p-values are to be understood in a descriptive manner. Missing values were not imputed, resulting in a complete case analysis.

## 3. Results

A total of 281 patients were included in the 6.5-year observation period from 2009 until June 2016 (Table 2). Descriptive analysis stratified by subtype is shown in Table 2. In all four subtypes, more than 50% of the patients were older than 70 years. The most frequent histologic subtype in the overall cohort was adenocarcinoma (47.0%). In 14 patients (5.0%) only an R1 resection was possible. A total of 159 patients (56.6%) were treated with adjuvant chemotherapy, irrespective of pT3N0 subtype. Three patients (1.1%) received postoperative radiotherapy, and adjuvant chemoradiotherapy was administered to 10 patients (3.6%).

The median follow-up time of the population was 81.0 months (95% confidence interval: (76.5–85.9); stability interval: (61.2, 101.8)). For the study population as a whole, median overall survival (OS) was not reached and 5-year OS was 66.7% (0.61, 0.72) (Figure 1a).

After stratification, according to the four major subtypes, the groups showed significant OS differences (Figure 1b). The group with tumor size >7 cm (5-year OS % 71.5 (0.62, 0.79)) had the best prognosis, and the mixed subtype (5-year OS % 50.3 (0.34, 0.65)) had the worst prognosis (Figure 1b and Table 3).

In an attempt to better understand these OS differences, we grouped the subtypes tumor larger than 7 cm and multiple nodules as group T3a, and the subtypes parietal pleural invasion and mixed as group T3b (Figure 1c). The OS rates were significantly different (*p* < 0.01) and revealed a 5-year OS of 71.3 % [0.64, 0.78] for T3a in contrast to a 5-year OS of 54.5 % (0.42, 0.66) for T3b.

OS survival stratified by delivery of adjuvant chemotherapy was evaluated as well. An overview on patient stratified by chemotherapy is presented in the Appendix A. For the pT3N0 cohort as a whole, adjuvant chemotherapy provided no survival benefit (Figure 2a). The same was true for the T3a group (Figure 2b). Only the T3b group suggested some benefit from adjuvant chemotherapy, although numbers here remain low (Figure 2c). Tested with a Cox model, we found a similar effect of adjuvant chemotherapy for the T3b group (Table 4). Although not significant (*p*-value = 0.068) the effect still points in the same direction with a rather low probability of error.

After adjustment, the OS difference from the mixed and parietal pleural invasion subgroup (T3b) to the larger 7 cm and multiple nodules subtype (T3a) remained significant in a Cox model (Table 5).

## 4. Discussion

pT3N0 lung tumors are a unique group in UICC Stage IIB of TNM-7 and TNM-8, characterized by local tumor progression without lymph node invasion. In 2016, the IASLC published the new TNM-8 classification [1]. Several recent validation studies highlight the prognostic value of TNM-8 superior to that of the 7th edition [13,14,15,16]. Our study highlights the urgent need for an improved understanding of the impact of the T descriptors on survival and the role of adjuvant chemotherapy.

Our data suggest that patients with tumors >7 cm or multiple nodules in one lobe have a significantly better survival than patients with pleural invasion or more than one pT3N0 descriptor (mixed subgroup). In 2015, the tumor descriptor >7 cm was reclassified as T4 in TNM-8 (Table 1). Our data provide an argument for a different grouping of the current T3 descriptors. First, one could argue that our results do not support migration of the T descriptor >7 cm to the T4 category in the TNM-8 classification. Second, patients with multiple nodules have a similar OS as patients with tumors >7 cm, supporting the notion that it should remain in the pT3 category, as in the current TNM-8 version and in other investigations [17]. Third, our data describe a novel mixed subtype that identifies a subgroup with a significantly worse survival within the pT3N0 cohort, suggesting that it could be moved to a more advanced category (either T3b or T4). This finding is supported by an analysis from the Netherlands, which also found that a mixed subtype was associated with worse survival compared to other T3 descriptors [18].

A total of 79% of the 22,257 patients included in the IASLC data came from Asia. Consequently, 440 of the patients with pT3N0 lung cancer (*n =* 2108) were non-Asian [2]. Therefore, our work with 281 patients from Europe should not be underestimated to discuss T-descriptor redefinition in a worldwide context. Apart from ethnicity, possible reasons for differences between our findings and those reported in the IASLC database could be of structural nature such as different distribution of histologic subtypes. In addition, details of surgical lymph node assessment and adjuvant treatment of patients in the IASLC database are not always reported.

Our data show important stratified long-term survival results after adjuvant chemotherapy. The current ASCO and ESMO guidelines recommend adjuvant chemotherapy for UICC IIB lung cancer [7,19]. According to current meta-analyses, adjuvant chemotherapy translates into a 4–5% absolute increase of 5-year survival [20,21]. This survival benefit, however, seems to be restricted to cases with lymph node involvement [22]. In cases without lymph node involvement, a pooled analysis of two randomized controlled trials of patients with T-size stratified cancers failed to show significant OS benefit depending on tumor size [23]. Other studies investigating pT3N0 subtypes are not comparable to our data since staging was based on TNM-6 and/or because T subtypes were not evaluated [7,24]. This demonstrates the need for prospective data from large adjuvant chemotherapy trials before guidelines can be updated to reflect changes in the current TNM classification. For now, specific data on the effect of adjuvant therapy for pT3N0 tumors are limited, and it is unclear how these patients should be treated. Our data suggest that adjuvant therapy does not improve long-term survival if the pT3N0 classification is based solely on the descriptors tumor >7 cm or presence of multiple nodules. Kaplan–Meier survival analyses do, however, suggest that patients with pleural invasion or mixed subtype may benefit from adjuvant chemotherapy. It seems reasonable that pleural invasion and mixed T3 descriptors are better markers for systemic tumor spread, therefore providing a better target for systemic therapy than pT3 tumors without extralobar progression. In summary, our work identifies a novel group of patients with locally advanced lung cancer tumors, who could particularly benefit from adjuvant therapy. This group could be an important target for further evaluation concerning prognosis and multimodal therapy approaches.

Overall, our patients had much better outcomes than those included in the Dutch and IASLC analyses, which raises the question of why this is (Table 3). We have to bear in mind that many of the databases used in the IASLC analysis were not specifically designed to study TNM classification. Tumor size was regularly recorded, but other T3 and T4 descriptors for defining T subcategories were frequently missing, which prevented the validation of these T3 and T4 descriptors [2]. As discussed before, ethnicity may play a role, as European patients were underrepresented in the IASLC analyses for TNM-8. Additionally, the Dutch study analyzed 683 patients, 294 of whom (43%) had unknown positive microscopic resection margins [18]. This is a much higher percentage than in our cohort, and positive resection margins could lead to worse OS. Our data provide information about censoring, but this information is not given in the Dutch and IASLC data. Finally, in terms of selection bias, our bicentric database cannot be directly compared with multi-institutional data. Nevertheless, up-to-date survival estimates for lung cancer suggest that survival rates in Germany could be higher, compared to other European countries [25].

Although this work is a not a bicentric analysis, the authors´ institutions combined represent one of the biggest surgical institutional collectives in Europe when it comes to interpreting IASLC data. Together, they represent 63.8% of the non-Asian or 13.3% of lung cancer cases documented in the last worldwide IASLC analysis. Thus, these data could play an important role in future redefinition of the T descriptor [2].

We cannot rule out a selection bias due to the retrospective character of our investigation. Our data are limited by the fact that the descriptor “multiple nodules” does not distinguish between synchronous multiple primary tumors and true intrapulmonary metastases (multiple Stage I tumors within the same lobe should have a better prognosis). Taken together, this proposal needs to be validated in the next IASLC staging project, in which the 9th edition of the TNM classification for lung cancer will be updated [26].

## 5. Conclusions

In summary, this study on NSCLC pT3N0 subtypes suggests that the T descriptor should be redefined. Tumors >7 cm should not be moved to pT4. Patients with a mixed subtype or pleural invasion have significantly worse survival than the other subgroups of pT3N0. Finally, patients with T3b tumors—in contrast to intralobar tumors without lymph node invasion—benefit from adjuvant therapy.

## Figures and Tables

**Figure 1 cancers-13-01812-f001:**
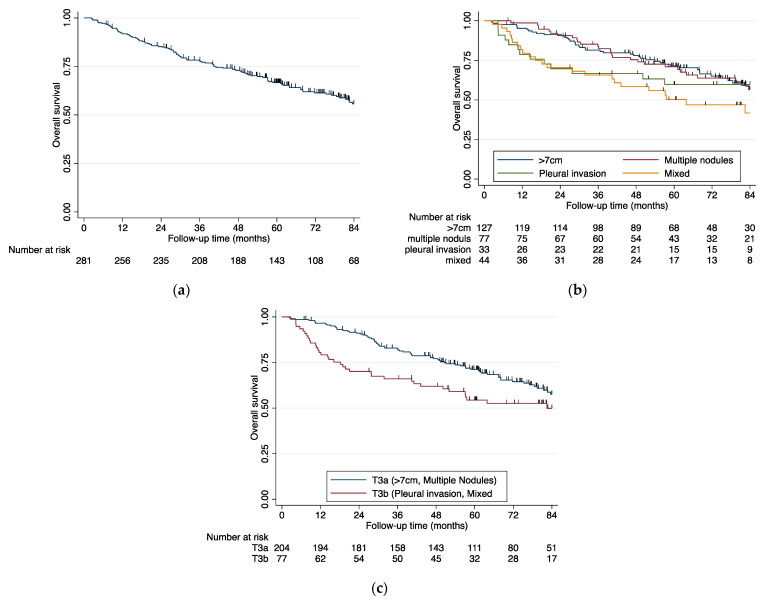
(**a**) Kaplan-Meier survival curve of the whole study population with pT3N0 lung cancer (**b**) Kaplan–Meier survival curves of the four major pT3N0 subtypes: tumor larger than 7 cm, second or multiple nodules in the same lobe, parietal pleural invasion, and the mixed subtype, logrank test *p =* 0.0085 (**c**) Kaplan–Meier survival curves of the pT3N0 lung cancer stratified for t3-descriptors grouped in T3a and T3b, logrank test *p =* 0.0026.

**Figure 2 cancers-13-01812-f002:**
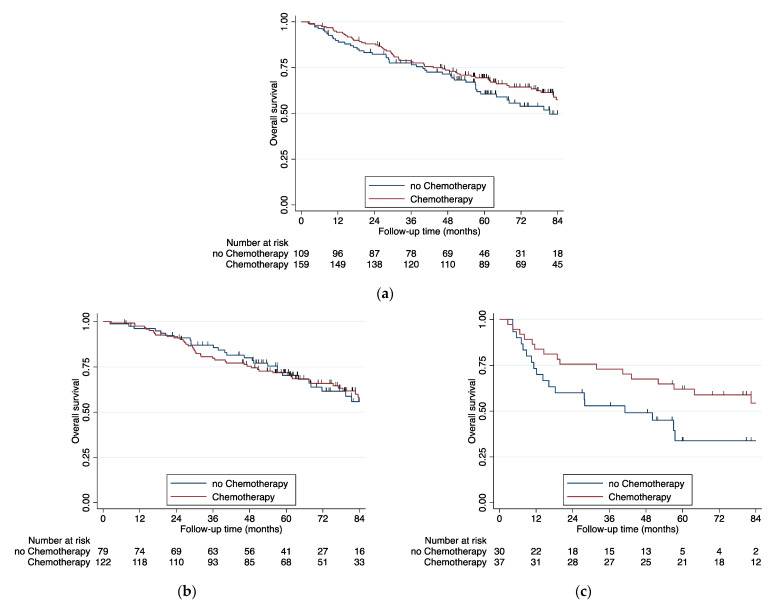
Benefit of adjuvant chemotherapy. Patients with adjuvant radio–chemotherapy/radiotherapy are excluded. (**a**) All patients, logrank test *p =* 0.2207 (**b**) tumor larger than 7cm and second or multiple nodules in the same lobe (T3a), logrank test *p =* 0.9880 (**c**) Parietal pleural invasion and the mixed subtype (T3b), logrank test *p =* 0.0261.

**Table 1 cancers-13-01812-t001:** Changes in NSCLC T3 descriptors between 6th, 7th, and 8th Edition.

T-Descriptor	TNM-6	TNM-7	TNM-8
Diameter 5–7 cm	T2	T2b	T3
<2 cm distance from the carina	T3	T3	T2
Complete atelectasis	T3	T3	T2
Separate tumor nodules in same lobe	T4	T3	T3
Parietal pleural/chest wall invasion	T3	T3	T3
Diameter >7 cm	T2	T3	T4
Diaphragm invasion	T3	T3	T4

**Table 2 cancers-13-01812-t002:** Descriptive analysis of the characteristics of the four major pT3N0 subtypes.

	>7 cm		Multiple Nodules		Parietal Pleural Invasion		Mixed		Total		
	No.	%	No.	%	No.	%	No.	%	No.	%	*p* Value
All	127	45.2	77	27.4	33	11.7	44	15.7	281	100.0	
Gender											0.231
Male	78	61.4%	41	53.2%	20	60.6%	31	70.5%	170	60.5%	
Female	49	38.6%	36	46.8%	13	39.4%	13	29.5%	111	39.5%	
ECOG											0.060
0	106	83.5%	58	75.3%	28	84.8%	39	88.6%	231	82.2%	
≥1	21	16.5%	19	24.7%	5	15.2%	5	11.4%	50	17.8%	
Age											0.602
<60	14	11.0%	6	7.8%	4	12.1%	8	18.2%	32	11.4%	
60–69	24	18.9%	19	24.7%	9	27.3%	9	20.5%	61	21.7%	
≥70	89	70.1%	52	67.5%	20	60.6%	27	61.4%	188	66.9%	
Surgery											<0.001
Lobectomy	106	83.5%	62	80.5%	32	97.0%	36	81.8%	236	84.0%	
Sleeve Resection	5	3.9%	1	1.3%	0	0.0%	0	0.0%	6	2.1%	
Sublobar Resection	0	0.0%	12	15.6%	1	3.0%	0	0.0%	13	4.6%	
Bilobectomy	8	6.3%	0	0.0%	0	0.0%	3	6.8%	11	3.9%	
Pneumonectomy	8	6.3%	2	2.6%	0	0.0%	5	11.4%	15	5.3%	
Resection side *											0.380
Right	74	58.3%	48	62.3%	22	66.7%	25	58.1%	169	60.4%	
Left	53	41.7%	29	37.7%	11	33.3%	18	41.9%	111	39.6%	
Residue											0.001
0	123	96.9%	77	100.0%	29	87.9%	38	86.4%	267	95.0%	
1	4	3.1%	0	0.0%	4	12.1%	6	13.6%	14	5.0%	
Histology											0.239
Adenocarcinoma	55	43.3%	42	54.5%	17	51.5%	18	40.9%	132	47.0%	
Squamous cell carcinoma	50	39.4%	31	40.3%	11	33.3%	20	45.5%	112	39.9%	
Other	22	17.3%	4	5.2%	5	15.2%	6	13.6%	37	13.2%	
Adjuvant therapy											0.001
No therapy	39	30.7%	40	51.9%	13	39.4%	17	38.6%	109	38.8%	
Chemotherapy	86	67.7%	36	46.8%	15	45.5%	22	50.0%	159	56.6%	
Radio–Chemotherapy	1	0.8%	1	1.3%	4	12.1%	4	9.1%	10	3.6%	
Radiotherapy	1	0.8%	0	0.0%	1	3.0%	1	2.3%	3	1.1%	

* One patient with missing side.

**Table 3 cancers-13-01812-t003:** Comparison of 5-y overall survival (OS) among subsets of patients with pT3N0 NSCLC (TNM-7).

	IASLC Database	Netherland Database	Heidelberg and Berlin Database
All pT3N0 (Stadium IIB)	56%	47.9%	66.7%
>7 cm	41%	47.2%	71.5%
multiple nodules	49%	62.8%	71.0%
parietal pleural invasion	49%	45.3%	59.8%
mixed	49%	28.7%	50.3%

**Table 4 cancers-13-01812-t004:** Cox Regression analysis of the T3b subgroup. Patients with adjuvant radio–chemotherapy/radiotherapy are excluded.

	HR	St. Err.	*p*-Value	(95% Conf.	Interval)
Age	1.05	0.02	0.021	1.01	1.09
Gender (RC: male)	0.61	0.22	0.185	0.29	0.28
ECOG >0 (RC: ECOG = 0)	0.87	0.51	0.752	0.27	2.80
Squamous-cell-C (RC: Adeno-C)	1.76	0.67	0.144	0.83	3.73
Other C (RC: Adeno-C)	2.13	1.06	0.124	0.81	5.58
CTX (RC: no CTX)	0.51	0.19	0.068	0.24	1.05

RC: reference category, C: Cancer, CTX: adjuvant chemotherapy.

**Table 5 cancers-13-01812-t005:** Cox regression analysis of patients with pT3N0 NSCLC. Patients with adjuvant radio–chemotherapy/radiotherapy are excluded.

	HR	St. Err.	*p*-Value	(95% Conf.	Interval)
T3b (RC: T3a)	2.34	0.48	<0.01	1.57	3.49
Age	1.06	0.01	<0.01	1.03	1.08
Gender (RC: male)	0.76	0.15	0.179	0.51	1.13
ECOG >0 (RC: ECOG = 0)	1.45	0.40	0.171	0.85	2.48
Squamous-cel-C (RC: Adeno-C)	1.17	0.25	0.463	0.78	1.79
Other CA (RC: Adeno-CA)	1.56	0.47	0.137	0.87	2.82
CTX (RC: no CTX)	0.92	0.20	0.708	0.60	1.41

RC: reference category, C: Cancer, CTX: adjuvant chemotherapy.

## Data Availability

Data available upon request due to privacy and ethical restrictions.

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
