# Peer review of "Validation of the T Descriptor (TNM-8) in T3N0 Non-Small-Cell Lung Cancer Patients; a Bicentric Cohort Analysis with Arguments for Redefinition"

_cancers, 2021, doi:10.3390/cancers13081812_

Round 1

Reviewer 1 Report

  1. Simple summary, line 22: “to optimize the definition of tumor classification” should be replaced with “to optimize tumor classification
  2. Simple summary, line 23 “and identify patients, who could benefit from an additional…” should be replaced with “and identify patients who could benefit the most from an additional
  3. Simple summary, line 24 “how a new developed redefinition of tumor…” should be replaced with “how a new redefinition of tumor classification
  4. Simple summary, line 25: the new classification could lead to a better stratificationof patients and a better definition of the adjuvant therapy more than improved survival outcomes.
  5. page 2, line 79: “staged as pT3N0 according to the…” should be replaced with “staged as pT3N0 lung canceraccording to the..”
  6. page 4, Table 2: it would be interesting to add the p-value when comparing the characteristics of the four major pT3N0 subtypes in the table, to better analyze differences among groups.
  7. page 5, line 143: Figure 1 has been labeled as Figure 1.
  8. page 5, line 144: “;” should be replaced by “,”
  9. page 5, line 150: there is repetition “% %”
  10. page 5, line 154: “((“ one parenthesis should be removed
  11. page 5, line 159: “71.3 [0.64,0.78] %” should be replaced with “71.3 % [0.64,0.78]”
  12. page 5, line 162: Cox-Model not Cox Modell
  13. page 6, lines 165 and 167: “Table 3a Cox Modell of patients with pT3N0 NSCLC” should be replaced with “Table 3a. Cox Regression Analysisof patients with pT3N0 NSCLC” and similarly “Table 3b Cox Modell of T3b patients with pT3N0 NSCLC” should be replaced with “Table 3b. Cox Regression Analysis of T3b patients with pT3N0 NSCLC”
  14. page 6, line 173: “application” should be replaced with “delivery”
  15. page 6, line 175: the dot between T3a group and (Figure 2a) should be removed.
  16. page 7, line 181: a dot should be added after Table 4.
  17. page 7, line 184: “lung cancer tumors” should be replaced with “lung tumors”.
  18. page 7, lines 186-188: “79% of the 22,257 patients included 186 came from Asia. Consequently 440 of the patients with pT3N0 lung cancer (n=2108) in this 187 database were non-Asian [2].”should be placed further down in the text (e.g. starting to line 203, before the explanation “apart from ethnicity…”)
  19. the discussion should be widened by citation of recent study on the topic (e.g. Jung HS, Lee JG, Lee CY, Kim DJ, Chung KY. Validation of the T descriptor in the new 8th TNM classification for non-small cell lung cancer. J Thorac Dis. 2018;10(1):162-167. doi:10.21037/jtd.2017.12.20)
  20. page 7, line 197: “second, the pT3 descriptor multiple nodules has a similar OS to tumors>7” should be replaced with “second, patients with multiple nodules has a similar OS to tumors>7”
  21. page 8, line 232: IASCL should replace ISLAC
  22. page 8, line 239: had unknown microscopic positive margin.
  23. page 8, line 245: this work is a bicentric analysis, is there an error?
  24. page 8, line 251: multiple nodules should be placed between inverted commas.
  25. page 8, line 257: “this study on NSCLC” not of
  26. page 8, line 261: “would be most likely to benefit from” be should be removed.

Reviewer 2 Report

This retrospective bi-centre cohort analysis included a decent number of surgical patients with pT3N0 NSCLC and primarily addresses the survival discrepancies based on the T3 descriptor reporting more favourable survival outcome in patients with tumor size > 7cm/multiple nodules vs. parietal pleura invasion/mixed. In addition, the receipt of adj. therapy was also addressed. The manuscript is well-written and provides a rationale for T-descriptor re-classification. An added merit is the number of patients approx. equalling 64%/13% of the Non-Asian/total pT3N0 cases in the last IASLC analysis. However, some issues need to be addressed before the manuscript can be considered for publication.

Major:

- In order to more extensively discern the effect of adj. therapy, more in-depth analysis is required: what was the histology, sex, age, ECOG-PS etc. of patients in the no therapy vs. chx subgroup as possibly patients with more adverse risk factors could inherently have received adj. therapy. Please perform additional analysis (analyse differences between groups)

Minor:

- Page 3, line 99: Please edit “…cases where criteria for two or more of the three major subtypes was present” to “…cases where criteria for two or more of the three major subtypes were present”

- Pls edit Cox Modell to Cox model throughout the manuscript

- Page 5, line 149-150, pls edit “For the study population as a whole, median OS was 96.7 [82.6, NA] months and 5-year OS was 66.7% % [0.61, 0.72]” to “For the study population as a whole, median OS was not reached [82.6, NA] months and 5-year OS was 66.7% % [0.61, 0.72]”

- Page 5; this should be figure 1 and not figure 2

- Table 3b please edit title to “Cox model of the pT3b subgroup”

- It makes more sense to include tables 3a/3b after figure 2 or modify figures 2a/b to 1e/f

- Page 6, lines 173-178: It would be more logical to include the paragraph “OS survival stratified by application of adjuvant chemotherapy was evaluated as well (Figure 1d)…Only the T3b group suggested some benefit from adjuvant chemotherapy, although numbers here remain low (Figure 2b)” after page 5 lines 156-160 “In an attempt to better understand these OS differences…for T3a in contrast to a 5-year OS of 54.5 % [0.42, 0.66] for T3b.” prior to addressing the cox proportional hazards models

- why weren’t the other covariates importantly gender included in the cox regression analysis?

- Page 8, line 232: Please edit “ISLAC” analyses” to “IASLC”

Round 2

Reviewer 2 Report

I have read with interest the revised manuscript. All my queries have been adequately addressed. I, therefore, endorse publication.